# Slow RNAPII Transcription Elongation Rate, Low Levels of RNAPII Pausing, and Elevated Histone H1 Content at Promoters Associate with Higher m6A Deposition on Nascent mRNAs

**DOI:** 10.3390/genes13091652

**Published:** 2022-09-14

**Authors:** Alicia Gallego, José Miguel Fernández-Justel, Sara Martín-Vírgala, Magdalena M. Maslon, María Gómez

**Affiliations:** 1Centro de Biología Molecular Severo Ochoa (CBMSO), Consejo Superior de Investigaciones Científicas/Universidad Autónoma de Madrid (CSIC/UAM), Nicolás Cabrera 1, 28049 Madrid, Spain; 2Center for Advanced Technology, Adam Mickiewicz University, Uniwersytetu Poznanskiego 10, 61-614 Poznań, Poland; 3Department of Gene Expression, Institute of Molecular Biology and Biotechnology, Adam Mickiewicz University, Uniwersytetu Poznanskiego 6, 61-614 Poznań, Poland

**Keywords:** histone H1, m6A, RNAPII elongation rate, RNAPII pausing, bivalent chromatin

## Abstract

N6-methyladenosine modification (m6A) fine-tunes RNA fate in a variety of ways, thus regulating multiple fundamental biological processes. m6A writers bind to chromatin and interact with RNA polymerase II (RNAPII) during transcription. To evaluate how the dynamics of the transcription process impact m6A deposition, we studied RNAPII elongation rates in mouse embryonic stem cells with altered chromatin configurations, due to reductions in linker histone H1 content. We found that genes transcribed at slow speed are preferentially methylated and display unique signatures at their promoter region, namely high levels of histone H1, together with marks of bivalent chromatin and low RNAPII pausing. They are also highly susceptible to m6A loss upon histone H1 reduction. These results indicate that RNAPII velocity links chromatin structure and the deposition of m6A, highlighting the intricate relationship between different regulatory layers on nascent mRNA molecules.

## 1. Introduction

N6-methyladenosine (m6A) is one of the most prevalent internal modifications on mammalian mRNA, regulating many aspects of its fate, including pre-mRNA splicing, alternative polyadenylation, mRNA decay, or translation, with fundamental implications in a variety of physiological processes. m6A is installed co-transcriptionally by a multi-subunit writer complex, composed of the methyltransferase-like 3 (METTL3), methyltransferase-like 14 (METTL14), Wilms-tumor-1-associating protein (WTAP), and the factors RBM15/RBM15B, VIRMA, ZC3H13, and HAKAI (reviewed in [1]). Members of the writer complex can be recruited to chromatin by transcription factors, chromatin associated proteins, and post-transcriptional modifications of histones, like H3K36me3, present at gene bodies of active genes [2,3,4].

Interestingly, recent work has shown that m6A also regulates the transcription process, per se. For example, METTL3 is recruited to promoters of highly expressed genes in *Drosophila* cells in a transcription-dependent manner where it regulates the release of RNA polymerase II (RNAPII) from its paused state [5]. Also consistent with a role on the regulation of transcription, METTL3 modifies nascent RNAs derived from promoters and enhancers, protecting them from premature transcription termination [6]. In addition, earlier work on human cells suggested an inverse relationship between transcription efficiency and m6A levels [7]. Together, these studies indicate that active transcription elongation can facilitate METTL3 binding and m6A deposition on nascent RNAs which, in turn, promotes the release of RNAPII from a paused state into productive elongation. However, direct evidence showing the impact of RNAPII elongation speed on the modification of nascent RNAs is missing. Here, we set out to examine this by determining transcription elongation rates in mouse embryonic stem cells (mESCs). We analyzed in parallel mESCs triple knock-out for three of the genes encoding for the linker histone H1 (H1-TKO; [8]), since we recently showed that histone H1 depletion reduces m6A levels at nascent coding and non-coding RNAs [9]. In particular, we aimed to address two questions: (i) whether the transcription elongation rate is predictive of m6A deposition levels on nascent RNAs and (ii) whether it is modulated by promoter chromatin structure. We identified slow-transcribed genes as a unique class of genes in terms of having a distinct chromatin configuration with high levels of histone H1, marks of bivalency at their promoter regions, and displaying low RNAPII pausing and high m6A levels on nascent transcripts. In addition, slow-transcribed mRNAs showed high sensitivity to histone H1 loss, leading to a strong reduction in m6A. Our findings further emphasize the role of m6A methylation in promoting transcription and highlight the crosstalk between chromatin structure and the deposition of this mark to regulate mRNA metabolism at specific gene classes.

## 2. Materials and Methods

### 2.1. Cell Culture

Mouse embryonic stem cells were grown in DMEM (Invitrogen; Waltham, MA, USA), supplemented with 10% fetal bovine serum (Gibco; Paisley, UK), 1 × non-essential amino acids (Gibco; Paisley, UK), 1 mM sodium pyruvate (Gibco; Paisley, UK), 2 mM L-glutamine (Gibco; Paisley, UK), 50 µM b-mercaptoethanol (Gibco; Paisley, UK), 10^3^ U/mL LIF (ESGRO; Darmstadt, Germany), 100 U/mL penicillin (Invitrogen; Waltham, MA, USA), and 100 µg/mL streptomycin (Invitrogen; Waltham, MA, USA), at 37 °C and 5% CO_2_.

### 2.2. 4sU-DRB Sequencing (DRB-TT Seq)

Nascent transcription labeling assays were carried out as previously described [10,11]. Briefly, cells growing at 80% confluence were treated with 100 µM of inhibitor 5,6-dichlorobenzimidazole 1-β-D-ribofuranoside (DRB) (Sigma; Fukushima, Japan) for 3 h. Next, 4-thiouridine (4sU) was added at a final concentration of 1 mM for 10 min before cell harvest. Cells were lysed directly on a plate with 5 mL of TRIzol (Invitrogen), total RNA was isolated following the manufacturer’s protocol, and sonicated by two pulses of 30 s in a Bioruptor instrument. A total of 100 μg of sonicated RNA per cell line was used for biotinylation and the purification of 4sU-labeled nascent RNAs. Biotinylation reactions consisted of total RNA and EZ-Link HPDP-Biotin dissolved in dimethylformamide (DMF) and were performed in labeling buffer (10 mM Tris pH 7.4, 1 mM EDTA) for 2 h with rotation at room temperature. Unbound Biotin-HPDP was removed by a chloroform/isoamylalcohol (24:1) extraction in MaXtract tubes (Qiagen; Hilden, Germany). RNA was precipitated with a 10th volume of 5 M NaCl and one volume of isopropanol. Following one wash in 80% ethanol, the RNA pellet was left to dry and was resuspended in 100 μL RNase-free water. Biotinylated RNA was purified using a μMacs Streptavidin kit, eluted twice using 100 mM DTT and recovered using an RNeasy MinElute Cleanup column (Qiagen), according to the instructions. cDNA libraries were prepared using a NEBNext Ultra Directional RNA Library Prep Kit (New England Biolabs; Ipswich, MA, USA), following the manufacturer’s instructions. Libraries were pooled and sequenced in 2 × 75 paired-end reads on a NovaSeq platform at Fundación Parque Científico de Madrid. Two libraries from independent biological replicates of the 0-, 5-, 15-, and 45-min time-points per cell type were first sequenced at low depth to determine the transcriptional waves upon DRB removal (9 × 10^6^ reads on average per library). Then, the 0- and 5-min time-point libraries were sequenced at a higher depth to calculate RNAPII elongation rates (70 × 10^6^ reads on average per library).

Reads were aligned to the mm10 reference genome using Bowtie2 [12] and standard parameters. Replicates from the same cell type and condition were pulled and BigWig files were generated with the *bamCoverage* function from deepTools [13] and loaded in the IGV browser for data visualization. Meta-gene profiles were performed using the HOMER suite [14] and the Tag Directories from the aligned BAM files of each sample were created using the *makeTagDirectory* function. Histograms were computed with *annotatePeaks.pl*, setting the following parameters: −*size 52000* −*hist 20* −*ghist* −*d*, and, as inputs, the generated tag directories and the mouse reference protein-coding transcriptome, including positions transcription start sites (TSS) − 2 kb and TSS + 50 kb. This reference transcriptome was obtained from RefSeq and curated to include the longest version of each transcript, resulting in a list of 20,621 protein-coding genes that was used in the rest of the analyses.

### 2.3. Calculation of Transcription Elongation Rates

In order to obtain the transcription elongation rate information, sequencing alignments were processed using Matlab and a previously published protocol [15,16], with the following adjustments. First, the mouse reference transcriptome list including 20,621 protein-coding genes was filtered to exclude genes with low coverage (i.e., <0.7 reads per gene at the first 20 kb from the TSS). Second, genes shorter than 20 kb were excluded. Finally, 6835 genes were included for rate calculation. Sequencing alignments were then normalized by their total read number and gene coverage was calculated for each gene. The *processesfind_boundary_4sUDRBseq.m* script [15] estimated the transcription read waves for each gene at the 5 min time-point samples, normalized by the 0 min control. Based on a genome-wide binned profile of the nascent RNAs and the information about their exon positions, the wave boundaries were determined. Finally, a linear fit of the boundary as a function of time after release was used to estimate the elongation speed in kb/min units. Genes with rates ≤ 0.5 kb/min were excluded to avoid potential artifacts due to low read coverages. In addition, genes with no data in one of the two cell types were removed. This resulted in robust elongation rates for 5351 genes expressed in both cell types. Three-quantile division in the set of 5351 genes was used to separate the data into slow, medium, and fast elongation rate categories, for both WT and H1-TKO mESCs.

### 2.4. Mononucleosomal DNA Sequencing (MN-Seq) Analyses

Subconfluent cells were washed twice with PBS and crosslinked by a 10 min incubation with PBS + 1% formaldehyde at room temperature. Crosslinking reactions were stopped through the addition of 125 mM glycine and additional rocking for 5 min. Cells were collected by scrapping in 2 mL ice-cold PBS and protease inhibitors (10 μM leupeptin, 100 μM PMSF, 1 μM pepstatin, 2 μg/mL aprotinin), collected by centrifugation at 180× *g* for 5 min at 4 °C, resuspended in homogenization buffer (10 mM Tris pH 7.4, 1 mM EDTA, 0.1 mM EGTA, 15 mM NaCl, 50 mM KCl, 0.15 mM spermine, 0.5 mM spermidine, 0.2% NP-40, 5% sucrose) at a concentration of 2 × 10^6^ cells/mL and incubated on ice for 3 min. Nuclei were then layered on top of a 3.5 mL sucrose cushion (homogenization buffer and 10% sucrose), and centrifuged for 20 min at 900× *g* and 4 °C. The nuclei pellet was resuspended in ice-cold wash buffer (10 mM Tris pH 7.4, 15 mM NaCl, 50 mM KCl, 0.15 mM spermine, 0.5 mM spermidine, 8.5% sucrose) at a concentration of 2 × 10^6^ nuclei/mL and divided in 1 mL aliquots. Two of these aliquots were treated with 0 and 600 units of MNase (Thermo Fisher; Waltham, MA, USA) after adding 1 mM CaCl_2_ for 6 min at 25 °C. Enzymatic reactions were stopped by adding 9 mM EDTA and 3.5 mM EGTA and placing the samples at 4 °C. The fragments corresponding to the mononucleosomal fraction (around 150 base pairs, bp) were purified from agarose gels using a Speedtools PCR Clean Up Kit (Biotools, Madrid, Spain) following the manufacturer’s instructions. Sequencing libraries were generated with the NEBNext DNA Library Prep Kit (New England Biolabs; Ipswich, MA, USA) and sequenced in 2 × 75 bp paired-end reads at Fundación Parque Científico de Madrid.

Paired-end reads were aligned to the mm10 reference genome using the bwa mem algorithm. Only fragments delimited by two paired reads mapping the same chromosome and separated by less than 250 bp were considered for the analysis; those fragments were trimmed 25 bp at both ends and used as input for the DANPOS software [17], which calculates and compares a positioning score based on the shape and the width of the distribution of the reads between different conditions. Briefly, two lists of genomic coordinates were generated: one with the less positioned (fuzzier), and another one with the more positioned (less fuzzy) nucleosomes, both in WT and H1-TKO cells. Nucleosome positioning was computed setting different *p*-value cut-offs (0.1 to 5 × 10^−5^), and the number of nucleosomes showing increased fuzziness was calculated and plotted in WT and H1-TKO. Nucleosome profiles were generated with NUCwave [18].

### 2.5. Chromatin Enriched RNA (CheRNA) Sequencing Analyses

CheRNA data was downloaded from Fernández-Justel et al. [9]. Reads were aligned to the mm10 reference genome and to the Luciferase coding sequence used as spike-in with Tophat2 [19] using standard parameters. Aligned BAM files were merged with the reference mouse transcriptome of 26,021 genes. Reads per gene were normalized by the total number of aligned sequences for each experiment and the transcript size. Read means were computed for each replicate and cell type.

### 2.6. RNAPII Chromatin Immunoprecipitation (ChIP) Sequencing Analyses

RNAPII ChIP data was downloaded from Fernández-Justel et al. [9]. Reads were aligned to the mouse mm10 and human hg19 reference genomes (since 1/20th of human chromatin was added prior immunoprecipitation as spike-in, see [9] for details) using the bwa mem algorithm, and merged with the reference mouse transcriptome of 26,021 genes. Two types of intersections were performed. A list of the genes, including coordinates at TSS + 500 bp and the terminal transcription start site (TTS), was used for calculating RNAPII occupancies in the gene bodies, whilst a list including coordinates at ±500 bp from the TSS was used for the study of RNAPII occupancies in the promoters. The ratio between mouse and human reads was calculated to correct the H1-TKO cells signal, according to the formula: (1)$$TKOs=\frac{TKOr\cdot mouseWTreads/humanWTreads}{mouseTKOreads/humanTKOreads}$$
where *TKOs* is the spike-in normalized RNAPII signal and *TKOr* is the total reads normalized RNAPII signal. Reads were normalized by this correction factor, the total number of aligned sequences for each experiment, and the transcript sizes. Means were computed for both replicates in each cell type. To obtain the RNAPII pausing index for each gene, we calculated the ratio between RNAPII occupancy in the promoter and in the gene body.

### 2.7. m6A-CheRNA Immunoprecipitation (MeChRIP) Sequencing Analyses

MeChRIP data was downloaded from Fernández-Justel et al. [9]. Reads were aligned to the mm10 reference genome using Tophat2 [19] with standard parameters. bedGraph files loaded in the IGV browser were generated with the *bedtool genomecov* function. Aligned BAM files were merged with the reference mouse transcriptome of 26,021 genes. Since the m6A mark has been mainly reported around the TTS, we used a modified version of the transcriptome list that included the TSS − 2 kb and the TTS + 2kb positions as coordinates. Sequencing reads were normalized by the total number of aligned sequences in each experiment. For the quantification of m6A methylation levels, the number of reads between the MeRIP samples and the CheRNA input were computed per transcript. After excluding genes with no data in one of the two replicates, means were finally calculated for both replicates in each cell type.

### 2.8. Statistical Analyses

Pairwise comparisons between cell types and/or elongation rate groups were performed in R using the Wilcoxon and Mann-Whitney test. Correlations among variables were calculated based on Spearman’s ρ or Pearson’s r correlation coefficients in R. Mean, median, and standard deviation information of the analyzed distributions are detailed in Appendix A.

### 2.9. GO-Term Analysis

GO-term enrichment analyses were performed using Panther v17.0 software [20]. −Log10 of the adjusted *p*-value was computed for biological processes and molecular functions and the 20 terms with the lowest *p*-values were selected.

### 2.10. m6A Peak-Calling

Peak-calling was performed using *macs2* according to the parameters set in [21]. Both biological replicates of each condition were merged, and normalization with the inputs was included in the peak-calling. 

### 2.11. Deposited Datasets

The datasets of DRB-TTseq and MNase-seq experiments generated in this work were deposited in GEO: GSE213270. Datasets from CheRNA, RNAPII-ChIP, and MeChRIP sequencing experiments in WT and H1-TKO mESCs were generated in our laboratory and deposited in GSE166426 [9]. H3K4me3, H3K27me3, and H3K9me3 ChIP-seq data in WT and H1-TKO mESCs were downloaded from GSE75426 [22]. WT mESCs histone variants H1d and H1c ChIP sequencing data were downloaded from GSE46134 [23].

All scripts used in this work are publicly available at the GitHub repository: https://github.com/aliciagallego/rates_m6A accessed on 31 July 2022.

## 3. Results

### 3.1. Reduction in Histone H1 Content Affects RNAPII Elongation Speeds at Slow and Fast Genes Differently

To measure the speed of mRNA synthesis, we performed time-course RNA sequencing of nascent transcripts using the reversible RNAPII inhibitor, DRB (DRB-TT seq; [16,24]; Figure 1A). DRB blocks the transition from initiation to elongation of transcription, thus upon drug release, RNAPII molecules move synchronously from TSS-proximal positions into the gene bodies, allowing the measuring of the progression of the transcriptional wave-front (Figure 1B). Elongation rates on the protein-coding transcriptome were calculated as previously described (see Methods and [11,15]), allowing the accurate determination of RNAPII elongation speed at 5351 genes common on both cell lines and biological replicates. On average, wave-front movement and transcription elongation velocities were comparable between cell types, with a median speed of 2.98 and 3.03 kb/min in WT and H1-TKO mESCs, respectively. Density plots of the elongation rates revealed a bimodal distribution in both cell types, determined by a small group of genes transcribed at slow speed (less than 2 kb/min), and a wider group of genes transcribed at fast speed (around 3.5 kb/min) (Figure 1C). Notably, H1-TKO cells displayed an increase in the number of slow-rate genes and a shift towards higher elongation speed in the group of fast-rate genes. Sorting genes into three quantiles of elongation velocities showed that lowering the histone H1 content on chromatin has a dual effect on RNAPII elongation rates: there are a higher number of genes transcribed at slow rates (Wilcoxon, *p*-value = 9.22 × 10^−10^), while medium and fast genes are transcribed significantly faster (Figure 1D) (Wilcoxon, *p*-value = 1.79 × 10^−70^ and *p*-value = 3.04 × 10^−65^, for medium and fast rate genes, respectively). The increased elongation rates of medium and fast genes occurring in H1-TKO mESCs might be related to the increased fuzziness of nucleosomes when linker histones become limiting on chromatin (Appendix A), although this aspect needs to be further investigated.

### 3.2. Slow-Transcribed mRNAs Display Low RNAPII Pausing and High m6A Levels, and Histone H1 Depletion Strongly Decreases Their m6A Deposition

To relate different parameters of the transcriptional cycle with the RNAPII elongation speed, we first integrated genome-wide datasets of RNAPII occupancy (RNAPII ChIP-seq) and chromatin-enriched RNA production (CheRNA-seq) recently generated by us in both cell types [9] (Figure 2A,B). We found that higher elongation rates correlated with higher RNAPII promoter densities and mRNA output on chromatin at each cell type (Figure 2C,D), although only slow-transcribed genes were significantly different from the other two speed categories (Wilcoxon, *p*-value = 1.76 × 10^−77^ and *p*-value = 9.97 × 10^−89^, for medium and fast rate genes RNAPII levels; and *p*-value = 2.54 × 10^−131^ and *p*-value = 6.04 × 10^−174^, for medium and fast rate genes CheRNA levels in WT cells, respectively). Moreover, slow-elongating genes displayed a lower pausing index (calculated as the ratio between RNAPII ChIP-seq RPKMs at the promoter [TSS ± 500 bp] and the gene body [from TSS + 500 bp to TTS]), than the other two groups (Figure 2E), indicative of limited RNAPII initiation-to-elongation pausing at promoters in this speed rate gene class (Wilcoxon, *p*-value = 1.36 × 10^−33^ and *p*-value = 2.03 × 10^−20^, for medium and fast rate genes in WT cells, respectively). Importantly, reduction in histone H1 content did not result in statistically significant differences in any of the above comparisons, suggesting that the relationship between low RNAPII pausing, slow elongation speed, and low nascent mRNA levels occur regardless of the chromatin alterations mediated by H1 depletion.

We then addressed whether co-transcriptional m6A deposition was associated with RNAPII elongation velocity by analyzing m6A levels on nascent transcripts, as determined by chromatin-enriched RNA m6A immunoprecipitation and subsequent RNA-seq analysis (MeChRIP; [9]; Figure 2A). No significant correlation was found between elongation rates and m6A levels when addressing the entire set of genes (Spearman’s ρ = −0.071, *p*-value = 3.7 × 10^−7^ for WT; and ρ = −0.03, *p*-value = 0.032, for H1-TKO, respectively), nor when analyzing a reduced dataset of elongation rate measurements derived from a related WT mESCs line (Pearson’s r = 0.028, *p*-value = 0.44; [11]). Thus, although there is a tendency of having reduced m6A levels in faster transcribed genes (more evident in WT cells), it is not a simple binary relationship.

We have previously reported that m6A levels on chromatin-RNAs were reduced in H1-TKO cells relative to their WT counterparts [9]. Interestingly, we found that the difference between cell types was significant in the subset of genes transcribed at slow and medium speed, but not at fast-transcribed genes (Figure 2F) (Wilcoxon, *p*-value = 9.3 × 10^−17^ and *p*-value = 7.91 × 10^−18^, for slow and medium genes, respectively). We conclude that slow-transcribed mRNAs display low RNAPII pausing and high m6A levels, and that decreasing histone H1 content in chromatin alters m6A deposition preferentially at this gene-rate class.

### 3.3. Promoters of Slow-Rate Genes Are Marked by High Levels of Histone H1 and H3K27me3

To investigate the chromatin architecture at the promoter regions of the genes transcribed at different rates, we analyzed available ChIP-seq datasets of histone post-translational modifications related to transcriptional regulation, including H3K4me3, H3K9me3, and H3K27me3, from WT and H1-TKO mESCs [22]. We also addressed the nucleosomal configuration based on MNase-seq profiles in both cell types (see Methods) (Figure 3A). As anticipated from the results presented in Figure 2, the variations in promoter architecture were related to the increasing RNAPII elongation rates (leading to higher mRNA output) at each cell type; medium and fast genes have higher levels of H3K4me3, a mark of active promoters, and a higher occupancy of the TSS+1 positioned nucleosome in both WT and H1-TKO cells. Conversely, slow-rate genes displayed comparatively higher levels of H3K27me3, a histone mark characteristic of poised, PRC2-regulated bivalent promoters [25], and reduced occupancy of the +1 nucleosome. The levels of the silencing mark H3K9me3 were slightly higher at the promoters of slow-transcribed genes. In agreement with this, GO-term analyses of the three gene-rate groups revealed that slow-transcribed genes were enriched in processes related to the regulation of cellular organization and development in both cell types, whereas medium- and fast-transcribed genes were enriched in general metabolic processes (Appendix A).

Since slow-transcribed mRNAs have higher m6A levels and also display the highest reduction upon histone H1 depletion, we hypothesized that their promoter regions will be marked by higher histone H1 occupancy in WT conditions. As predicted, we found that slow-transcribed genes displayed higher levels of the histone variants H1c and H1d around their TSS compared to medium and fast-transcribed genes (Figure 3B) (Wilcoxon, *p*-value = 2.2 × 10^−18^ and *p*-value = 2.27 × 10^−17^ for the slow-medium and slow-fast comparison in H1c; and *p*-value = 2.35 × 10^−12^ and *p*-value = 1.22 × 10^−11^ for the slow-medium and slow-fast comparison in H1d, respectively). To relate histone H1 content and m6A reductions upon H1 depletion more directly, we sorted slow-transcribed genes by histone H1 promoter-proximal occupancies in WT cells and plotted the extent of m6A loss (MeChRIP H1-TKO/WT ratio) at the 684 genes common in both cell types (Figure 3C). Interestingly, we found that the largest decreases in m6A occur at slow genes with the lowest histone H1 occupancy within this group, suggesting that slow-elongating genes are highly dependent on appropriate histone H1 levels to ensure correct m6A deposition.

Finally, we tested whether, not only the total m6A levels, but also the relative distribution of m6A peaks along the gene bodies, vary between the three gene-rate groups (Figure 3D). We found that slow-transcribed genes showed a slight, although not significant, tendency to display m6A peaks around the TSS, relative to the medium- and fast-rate categories.

Overall, these integrative analyses suggest that the chromatin configuration imparted by high levels of histone H1 and H3K27me3/H3K4me3 at TSS-proximal regions leads to reduced RNAPII promoter-pausing and a slow elongation rate, together facilitating m6A deposition on nascent mRNAs encoding proteins critical for cellular and developmental processes.

## 4. Discussion

Pioneer studies by Slobodin and co-workers demonstrated that the deposition of m6A on mRNAs was linked to RNAPII transcriptional dynamics, which, in turn, affected mRNA translation. In particular, using RNAPII elongation mutants, they showed that a suboptimal rate of transcription resulted in higher m6A deposition on mRNAs and a tendency of lower translation efficiency [7]. Our results corroborate these findings and further show that, in physiological conditions, slow RNAPII transcription indeed imprints a considerable proportion of mRNAs. We found that around one third of the genes analyzed in mESCs are transcribed at less than 2 kb/min and are preferentially methylated. Since this gene set is enriched in functions relating to cellular and developmental processes, these observations suggest that slow transcription rates are registered in the form of higher m6A modification levels at developmental genes, likely contributing to keeping these key genes untranslated during pluripotent stages.

In addition, slow-rate genes show limited RNAPII pausing at their promoter regions, which is in agreement with the finding that m6A RNA modification has a positive effect on RNAPII pause release in *Drosophila* cells [5]. Their promoter regions display characteristics of a bivalent chromatin state, exhibiting both activating (H3K4me3) and silencing (H3K27me3) histone marks typical of many developmental genes on mESCs [25]. Interestingly, a large computational study addressing chromatin marks and RNAPII occupancy across multiple cell types found that bivalent genes in mESCs show low RNAPII pausing [26]. Altogether, these observations suggest that the reduced degree of RNAPII pausing at bivalent genes could be related to the elevated m6A deposition at this mRNA class.

A surprising finding from our work is that slow RNAPII-transcribed genes are highly susceptible to m6A loss upon histone H1 reduction. Consistently, these genes have comparatively higher levels of linker histone H1 at their TSS than the other speed groups. It follows that the chromatin composition of bivalent promoters is marked by high levels of histone H1, which likely contributes to its repressed state in mESCs. In a recent study, we unveiled a novel role of histone H1 in regulating non-protein coding RNA transcription and turnover on chromatin in a m6A-mediated manner [9]. Taking all of this evidence into account, we propose that histone H1 is an epigenetic regulator that mediates m6A modification on nascent RNAs that need to be restricted in pluripotent cells.

## Figures and Tables

**Figure 1 genes-13-01652-f001:**
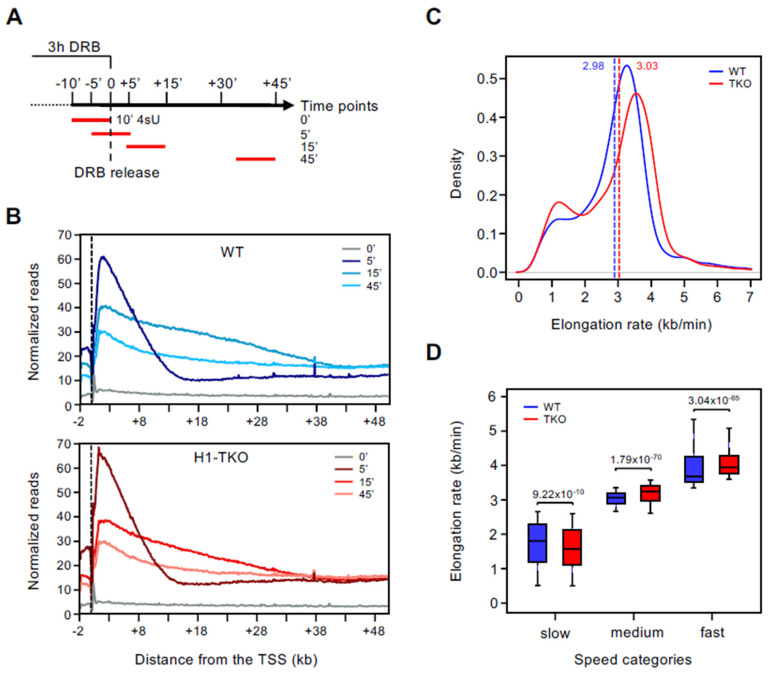
Determination of RNAPII elongation rates by DRB-TTseq. (**A**) Schematics of the DRB-TTseq approach used to calculate RNAPII elongation rates. The red lines denote the 10 min 4sU pulse immediately before cell harvest. (**B**) Profiles of normalized TTseq reads at the indicated time points within −2 kb/+50 kb window around TSSs of all coding gene promoters. Upper plot, WT cells; lower plot, H1-TKO mESCs cells. The dashed vertical line marks the TSS. (**C**) Density plots of elongation rate (kb/min) calculated for genes common in both cell lines (n = 5351). Median values of elongation rates for each cell line are indicated by vertical dashed lines and noted on top. Blue, WT; red, H1-TKO mESCs. (**D**) Box plots of elongation rates (kb/min) for the slow, medium, and fast elongation rate gene groups in WT (blue) and H1-TKO (red) mESCs. Differences between cell types were assessed by Wilcoxon test; only significant comparisons are shown. Medians are shown as horizontal lines. Outliers were not included. Total number of genes in WT: 1762 (slow), 1845 (medium), and 1744 (fast); and in H1-TKO: 1767 (slow), 1838 (medium), and 1746 (fast).

**Figure 2 genes-13-01652-f002:**
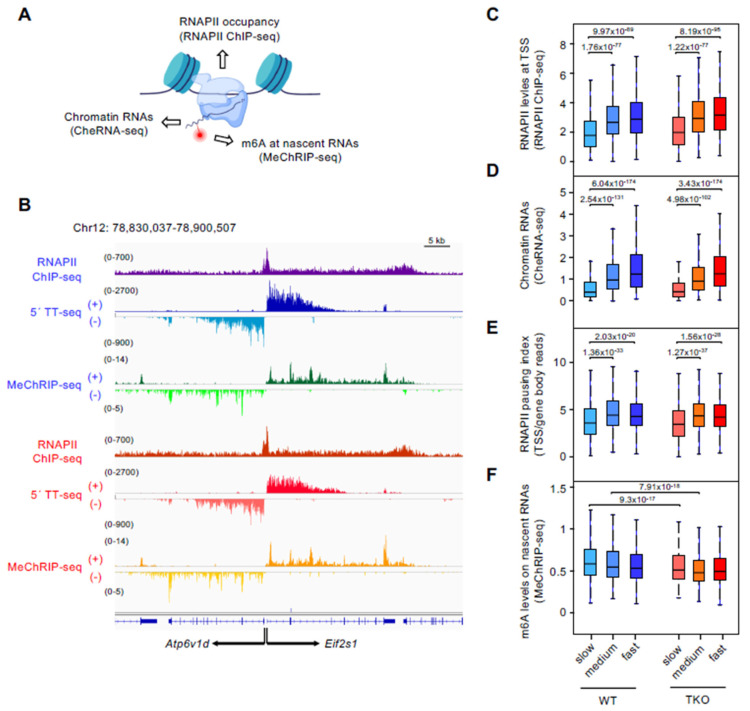
RNAPII dynamics, RNA output and m6A levels at the three identified elongation rate gene groups. (**A**) Schematics illustrating the genome-wide datasets derived from WT and H1-TKO mESCs analyzed at each elongation rate gene category. Data from Fernández-Justel et al. (2022). (**B**) Representative IGV genome browser tracks of RNAPII occupancies (RNAPII ChIP-seq), the wave of nascent transcription (5 min TTseq), and m6A levels at chromatin RNAs (MeChRIP-seq) in WT (first three rows, blue) and H1-TKO (last three rows, red) mESCs across the divergent *Atp6v1d* and *Eif2s1* genes. (**C**–**F**) Box plots showing the slow, medium, and fast elongation rate groups of genes in WT (blue) and H1-TKO (red) mESCs and their levels of RNAPII around the TSS (±500 bp) (RNAPII ChIP-seq) (**C**), chromatin-enriched RNAs (CheRNA-seq) (**D**), RNAPII pausing index (**E**), and m6A levels on chromatin RNAs (MeChRIP-seq) (**F**). Wilcoxon test: only significant comparisons are shown. Box plot annotations are as in Figure 1D.

**Figure 3 genes-13-01652-f003:**
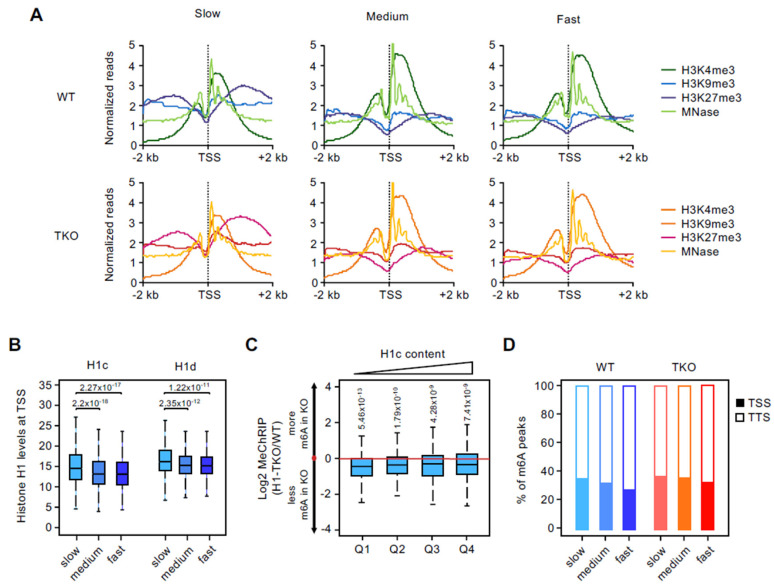
Promoter architecture at different elongation rate gene groups. (**A**) Profile of ChIP-seq signal of the indicated epigenetic marks and positioned nucleosomes plotted in the 4 kb window surrounding the TSS (±2 kb) of the studied elongation rate gene groups. Upper plots, WT; lower plots, H1-TKO mESCs. ChIP-seq data from Geeven et al. (2015). (**B**) Levels of histone H1 variants 1c and 1d around the TSS (±2 kb) at slow, medium, and fast elongation rate groups of genes in WT mESCs. H1 ChIP-seq data from Cao et al. (2013). (**C**) Ratios of normalized MeChRIP read counts relative to input between H1-TKO and WT mESCs (log_2_ H1-TKO/WT) across four quantiles of increased histone H1 levels. Only slow-genes common between both cell types were analyzed (n = 684). *p* values are shown on top (one-sample Wilcoxon rank-sum test). (**D**) Distribution of m6A peaks in the 4 kb window surrounding the TSS (±2 kb, solid bars) or TTS (±2 kb, blanked bars) at the indicated elongation rate groups.

## Data Availability

DRB-TTseq and MNase-seq datasets generated and analyzed in this work were deposited in GEO with the accession number GSE213270. CheRNA, RNAPII-ChIP, and MeChRIP sequencing datasets were generated in our laboratory and deposited in GSE166426 [9]. H3K4me3, H3K27me3, and H3K9me3 ChIP-seq datasets were downloaded from GSE75426 [22]. Histone variants H1d and H1c ChIP-seq datasets were downloaded from GSE46134 [23].

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
