# Peer review of "Slow RNAPII Transcription Elongation Rate, Low Levels of RNAPII Pausing, and Elevated Histone H1 Content at Promoters Associate with Higher m6A Deposition on Nascent mRNAs"

_genes, 2022, doi:10.3390/genes13091652_

Round 1

Reviewer 1 Report

This study tackles a very active area of research, about the interplay between the epitranscriptome and the epigenome and how this impacts the dynamics of the RNAPII complex. The key claim of the study is that m6A, on nascent RNA, and variants of the H1 histone, on chromatin, contribute shaping the transcriptional dynamics of specific gene sets. The study nicely integrates various publicly available datasets and generates few additional omics data. I appreciated the release of the code for reproducing the bioinformatic analyses. The major limitation of the study is that the key claim is mostly based on correlative analyses and small effect size. For example, the reduction in m6A is significative in Fig. 2F, mostly due to the large number of data points, yet the magnitude of the difference is minimal. A number of comments are listed below, suggesting clarifications and few additional analyses.

Major:

1.     The authors rely on DRB block and release experiments to quantify RNAPII elongation rates. This method was developed to overcome key limitations of RNAPII profiling by ChIP-seq. It is therefore unclear why RNAPII ChIP-seq data should be used to corroborate DRB block and release data. In Section 3.2: “As expected, higher elongation rates correlated with higher RNAPII promoter densities and mRNA output on chromatin at each cell type”. I do not see how faster RNAPII would imply higher density. For example, also stalled (i.e. not moving) RNAPII would imply very high RNAPII density. At the best, RNAPII density could help in measuring the stalling index, a very indirect measurement of pause-release dynamics.

2.     It is not clear what the contribution of the analyses presented in Fig. 3A is, since those data are never directly integrated with the level and changes of H1 variants and m6A.

3.     Since the association between m6A, elongation rates and level of H1 variants is the key point of this study, it would be nice to address it more directly. Indeed, the boxplots have quite a large spread, and it is not clear for example if the genes having the highest values of H1 variants are also those whose RNAs have the highest m6A levels and the greater m6A drop in H1-TKO cells.

4.     The notion that m6A is associated to higher rates of pause-release and lower rates of elongation is puzzling. It seems contradictory that RNAPII are more easily released from pausing state at TSS but are then slowed down during elongation. Indeed, it remains unclear to me when the authors recapitulate this as: “the transition of RNAPII from paused state into productive elongation at bivalent genes could be mediated through elevated m6A deposition” at line 352. How would this be in agreement with the notion of reduced elongation rates?

Minor

5.     Details on the origin and analysis of m6A data (Section 2.7) should be revised and clarified. MACS2 should not be used for m6A peak calling. More appropriate peak callers were developed, such as exomePeak and TRES.

6.     Omics data have to be properly deposited (“GSE ..........” in Section 2.11).

7.     The proper identification of the ends of RNAPII-released waves is critical for the estimation of RNAPII elongation rates. The authors should provide more details on how these end points were identified.

8.     Various filter on gene lengths were implemented in the original DRB block and release experiments, to ensure a proper identification of the advancing wave of released RNAPII. Have the authors followed those guidelines?

9.     It would be useful to report the number of genes for each group in Fig 1D.

10.  Line 263: is correlation exactly -0.071 for both WT and H1-TKO conditions? Also, the corresponding figures should be shown.

11.  Line 266: “there is a tendency of reduced m6A levels in fast transcribed genes (more evident 266 in WT cells)”. I guess this refers to Fig. 2F. The figure should be referenced in the text. In addition, a trend-test should be used to properly check for this tendency.

12.  The 2.7 section of the methods should be revised, it refers to RNAPII ChIP-seq data.

13.  Section 2.6. The rationale of aligning RNAPII ChIP-seq data to both human and mouse, and their joint normalization is unclear to me.

14.  The authors should discuss why would H1 loss increase elongation rates for faster genes.

15.  It is unclear what the contribution of the METTL3 binding data is, and it should be discussed why TTS regions are those with the highest fraction of m6A peaks but the lowest level of METTL3 binding.

Reviewer 2 Report

The Authors, to identify direct evidence showing the impact of RNAPII elongation speed on the modification of nascent RNAs, determined transcription elongation rates in mouse embryonic stem cells (mESCs) in 3 knock-out models for three of the genes encoding for the linker histone H1, because histone H1 depletion reduces m6A levels at nascent coding and non-coding RNAs.

According to their findings, 1) slow-transcribed genes show a distinct chromatin configuration with high levels of histone H1, marks of bivalency at their promoter regions, and low RNAPII pausing and high m6A levels on nascent transcripts; 2) slow-transcribed mRNAs showed high sensitivity to histone H1 loss, leading to strong reduction in m6A.

1) Fig 1D, Fig 2C/D/E/F, Fig. 3B/C: please insert the exact pvalues; please also add the sample size at the bottom of each boxplot; 2) Fig 3D is not clear, probably due to the low resolution.  3) All these figures should be supported by Tables reporting the exact number of genes for each class, and details of the distributions (i.e. mean + SD). 4) Any statement along the text should state the exact values: just to give an example in line 300, 'The levels of the silencing mark H3K9me3 were slightly higher at the promoters of slow-transcribed genes', it would be appropriate to mention in brackets the exact values that lead to this conclusion. A modification of this kind would make the text more comprehensible.

Please also check the link to the GitHub repository:  https://github.com/aliciagallego/rates_m6A points to https://github.com/aliciagallego/tfm_aliciagallego

Round 2

Reviewer 2 Report

The authors responded to the reviewer's requests.